# Telehealth Integration into Pharmacy Practice Curricula: An Exploratory Survey of Faculty Perception

**DOI:** 10.3390/pharmacy11040110

**Published:** 2023-06-30

**Authors:** Jennifer M. Bingham, David R. Axon

**Affiliations:** 1Department of Pharmacy Practice & Science, University of Arizona R. Ken Coit College of Pharmacy, Tucson, AZ 85721, USA; draxon@arizona.edu; 2Center for Health Outcomes and Pharmacoeconomic Research, University of Arizona, Tucson, AZ 85721, USA

**Keywords:** telehealth, digital health, pharmacy education, emerging technology

## Abstract

(1) Background: The use of telehealth in the United States during the coronavirus disease 2019 pandemic was accelerated and there was a lack of telehealth training programs available to clinicians of all levels. At the onset of the pandemic, the American Association of Colleges of Pharmacy (AACP) had no educational outcomes or professional activity standards for the inclusion of telehealth in the didactic Doctor of Pharmacy curriculum. Yet, in November 2022, the AACP encouraged colleges of pharmacy to include digital health and telehealth. The purpose of this study was to assess faculty perceptions in preparation for a nation-wide survey regarding telehealth integration into pharmacy practice curricula. (2) Methods: An exploratory questionnaire was developed to describe faculty perceptions and opinions of telehealth integration into the pharmacy practice curriculum at a single college of pharmacy. The questionnaire was emailed to 76 faculty members over six weeks in Summer 2022. Data were summarized descriptively. (3) Results: A total of 18 faculty members completed the survey (24% response rate). The responding faculty were typically very aware (median = 4) of telehealth, its benefits, and barriers, and were very comfortable (median = 4) discussing telehealth communication, benefits of telehealth, and barriers of telehealth. Yet, they were less comfortable discussing telehealth applications (median = 2.5). The faculty had a positive perception of telehealth in general (mean = 8.1 ± 1.5), telehealth services (mean = 8.6 ± 1.6), and the incorporation of telehealth instruction into the pharmacy practice curriculum (mean = 7.7 ± 2.7). Most respondents (67%) could discuss telehealth in their course. Lack of time to teach (50%) was the most reported reason by those who did not have plans to incorporate telehealth instruction into their course. (5) Conclusions: This exploratory survey of faculty at one college of pharmacy indicated positive perceptions and opinions of telehealth integration into the Doctor of Pharmacy curriculum. Further efforts to incorporate telehealth into the curriculum at other pharmacy schools is warranted.

## 1. Introduction

The coronavirus disease 2019 (COVID-19) pandemic underscored an unequitable digital divide [1]. As a digital health strategy, telehealth is defined by the Health Resources and Services Administration as the “use of digital technologies to communicate at a distance virtually, with computers and mobile devices, to access and manage health care services remotely” [2]. Although telehealth as a modality was heavily deployed in the pandemic, it presented challenges for healthcare providers to preemptively prepare for virtual provision of care [1]. Telehealth education as a strategy to enhance telehealth readiness remains crucial [1], especially as the demand for virtual care is predicted to increase in the foreseeable future with long-term adoption [3]. Some pharmacy programs incorporated tele-education rapidly into their curricula in response to the COVID-19 pandemic [3]. However, these curricula changes were often completed out of necessity to implement social distancing without the option for advanced preparedness. Furthermore, Frenzel et al. proposed that pharmacy programs should reflect this change in education [3].

Despite pharmacists’ positive patient contributions on the telehealth care team [4,5], there remains a deficit in telehealth integration into didactic education within pharmacy curricula [6]. Compared to professional pharmacy programs, there was also a dearth in medical school curricula of the incorporation of telehealth instruction into didactic education prior to the pandemic [7]. Chike-Harris et al. compared different healthcare training programs and found significant discrepancies in the consistency and depth of telehealth content between curricula [6]. Despite the need to standardize telehealth curricula and establish national competencies [8], Papanagnou et al. identified a lack of training available in North America to prepare clinicians of all levels for the modality of telehealth [9]. As a result, didactic and experiential learning through virtual interactions and the incorporation of telehealth is encouraged to address this disparity in curricula [1,3].

The American Association of Colleges of Pharmacy (AACP) develops educational outcomes to facilitate planning, delivery, and assessment of professional pharmacy program curricula [10]. At the onset of the pandemic, colleges of pharmacy were not expected to incorporate telehealth education into the curricula, as it was not a required AACP educational outcome [6]. Given the evolution of digital health technology and circumstances from the pandemic, the AACP sought expert opinion and developed a task force in November 2022 to update the standardized outcomes and professional activities [11]. As a result, it was recommended to incorporate topics related to digital health, including telehealth [11,12].

Research is warranted to determine the breadth and depth of how telehealth instruction is provided to student pharmacists in required coursework. Results of a survey may guide colleges of pharmacy as they create telehealth competencies within existing curricula to meet new guidelines set forth by the AACP. Given these changes and the emergence of digital technology in pharmacy practice, an exploratory survey was developed to assess perceptions at a single college of pharmacy in preparation for a nationwide survey. The study objective was to explore faculty perceptions and opinions of telehealth education integration within the professional Doctor of Pharmacy didactic curriculum.

## 2. Materials and Methods

### 2.1. Study Design, Setting, and Subjects

This was an exploratory study to determine the general perception of telehealth and opinions regarding the incorporation of telehealth education into Doctor of Pharmacy (PharmD) curricula. Faculty members employed at an accredited college of pharmacy in Arizona who taught in the pharmacy curriculum were eligible to participate in the study. Subjects were invited to participate in the survey regardless of expertise or practice experience. The Institutional Review Board approved this study (protocol 00001280 approved 10 June 2022).

### 2.2. Exploratory Questionnaire

The survey was created following a literature search to identify survey studies published in major pharmacy journals and extract relevant information. To identify studies, the search terms “pharmacy education,” “curricula,” and “PharmD” were combined. Next, two researchers extracted potential items from a published survey to be included in the questionnaire [13]. The published survey was selected because it questioned faculty regarding level of comfort, awareness, and perceptions teaching a novel topic in the classroom. Lastly, four focus areas were selected because of their relevance to telehealth readiness reported in the literature [1]. The final questionnaire consisted of 24 items and was developed into 4 focus areas, including awareness, perception, level of comfort, and instruction. Five demographic variables included sex, length of time as a faculty member at any college/school of pharmacy, length of time as a faculty member at the College of Pharmacy, whether the respondent had received any telehealth service in the past, and whether the respondent had a pharmacy degree. Two open-ended response questions were included to collect feedback comments not captured by the main questionnaire items (“Do you have additional thoughts or comments about the incorporation of telehealth into PharmD curricula?” and “Do you have any additional thoughts or comments about the importance of telehealth into clinical practice?”). 

#### 2.2.1. Awareness Questionnaire

The awareness questionnaire was measured using 4-point ordered response items and included (1) “How aware are you about telehealth?,” (2) “How aware are you about the applications of telehealth?,” (3) “How aware are you about the benefits of telehealth?,” and (4) “How aware are you about the barriers of telehealth?” Response options were “Not aware,” “Somewhat aware,” “Aware,” and “Very aware.”

#### 2.2.2. Perception Questionnaire

The perception questionnaire was measured using 10-point ordered response items and included (1) “How positive is your perception of telehealth in general?” (response options anchored at 0 = not very positive and 10 = extremely positive), (2) “How much value do you see in telehealth services?” (response options anchored at 0 = not very valuable and 10 = extremely valuable), and (3) “How much value do you see in the incorporation of telehealth instruction in the pharmacy practice curriculum?” (response options anchored at 0 = not very valuable and 10 = extremely valuable). 

#### 2.2.3. Level of Comfort Questionnaire

The level of comfort questionnaire was measured using 4-point ordered response items and included (1) “How comfortable do you feel discussing telehealth communication?,” (2) “How comfortable do you feel discussing telehealth applications?,” (3) “How comfortable do you feel discussing the benefits of telehealth?,” and (4) “How comfortable do you feel discussing telehealth barriers?” Response options were “Not at all comfortable,” “Somewhat comfortable,” “Comfortable,” and “Very comfortable.”

#### 2.2.4. Telehealth Instruction Questionnaire

The telehealth instruction questionnaire included (1) “Do you teach a subject area in which you can discuss telehealth?” (response options: “Yes,” “No”), (2) “Do you plan or have you already incorporated telehealth instruction into your course?” (response options: “Yes,” “No”), and (3) “What is the reason why you do not teach telehealth in your course?” (response options: “It is not relevant to my class/subject area,” “I do not think it is important,” “I do not know much about it,” “I think it is covered in another course,” “I do not have time to teach it,” “Other”). 

### 2.3. Data Collection and Analysis

Faculty members were invited to voluntarily participate in this study via email. Data were collected through REDCap^®^ (Research Electronic Data Capture, Vanderbilt University, Nashville, Tennessee) [14]. The survey was emailed to 76 faculty members in the Doctor of Pharmacy program at a single college of pharmacy in summer 2022. The initial email was sent in mid-June, with reminder emails sent in mid-July and mid-August. Data collection stopped at the end of August 2022. Data were analyzed using Microsoft Excel (v.16.73, Redmond, WA, USA) as summary statistics, which included frequencies and percentages for nominal data, medians with interquartile ranges for ordinal data, and means with standard deviations for continuous data.

## 3. Results

A total of 18 faculty responded to the survey, resulting in a 24% response rate. Overall, 8 (47%) respondents had served as faculty at any college/school of pharmacy for 11–20 years and 7 (41.2%) had served at the College of Pharmacy for less than 5 years. Most of the respondents had a pharmacy degree (82.3%), indicated female sex (70.6%), or had received a telehealth service (67%). See Table 1.

A median response of “very aware” was found for respondent awareness about telehealth (median = 4, IQR = 0.8), its benefits (median = 4, IQR = 0), and barriers (median = 4, IQR = 1).

A mean response score of 8.1 (SD ± 1.5) out of 10 was found for positive faculty perception of telehealth in general. A mean score of 8.6 (SD ± 1.6) out of 10 was found for value seen in telehealth services and 7.7 (SD ± 2.7) out of 10 for value seen in the incorporation of telehealth instruction in the pharmacy practice curriculum. 

In addition, faculty reported being very comfortable discussing telehealth communication, benefits of telehealth, and barriers of telehealth (median scores = 4). However, the faculty level of comfort discussing telehealth applications was reported as being between somewhat comfortable and comfortable (median = 2.5, IQR = 2.0).

Most respondents (67%) reported that they taught a subject area in which they could discuss telehealth. Lack of time to teach (50%) was the most common reason reported by those who were not, or did not, have plans to incorporate telehealth instruction into their course. See Table 2.

Six responding faculty members offered comments about the incorporation of telehealth into the curriculum, and three offered comments about the importance of telehealth in clinical practice. See Appendix A—Table A1.

## 4. Discussion

This exploratory study assessed pharmacy faculty perceptions, awareness, and level of comfort integrating telehealth into existing curricula. 

The data for length of time as a faculty member were well distributed between all three response options, including relatively new faculty members with less than 5 years of experience, more established faculty members with 6 to 10 years of experience, and senior faculty members with 11 to 20 years of experience. The majority of respondents possessed a pharmacy degree, which is perhaps unsurprising given that the focus of this study was on clinical telehealth services. Most respondents were female, which is greater than their proportional representation within the college.

The survey included questions to assess the general perception of telehealth from the faculty. Overall, perceptions of the value of telehealth services and incorporation into pharmacy practice curricula were positive. These findings align with another study that found positive healthcare provider perception of telehealth as it relates to quality and care delivery [15]. Next, the survey aimed to identify faculty awareness of telehealth. Faculty reported feeling very aware of telehealth and its benefits and barriers. As the COVID-19 pandemic necessitated a rapid increase in the use of telehealth as a modality of care [16], several barriers to effective implementation and optimization still exist. Although there are multiple benefits to the use of telehealth [16], intentional integration of telehealth into pharmacy practice curricula will allow student pharmacists to preemptively train to overcome implementation barriers as pharmacists. 

Then, the survey explored faculty level of comfort teaching telehealth, including its benefits, barriers, and different applications. Although most were comfortable discussing telehealth, it was found that there was a lower level of comfort from the faculty with discussing different telehealth applications. Interestingly, Frenzel et al. proposed that tele-education programs should improve understanding of telehealth technologies, etiquette, governing practices, and reimbursement [3]. Although our survey did not capture participants’ formal training on telehealth communication or digital health, our findings support the observed need for improved didactic education and simulation as a strategy to improve the understanding of telehealth [17]. The Association of American Medical Colleges provides a series focused on telehealth competencies to guide teaching and learning that could be adapted for pharmacy education [18]. Although continuous professional development programs exist in pharmacy, such as the American Society of Health-System Pharmacists Telehealth Certificate program for pharmacists and pharmacy technicians [19], our study highlights the need to develop telehealth competencies specific to pharmacy practice curricula. 

Lastly, the survey included questions to describe reasons as to why telehealth instruction was not integrated into their courses. Most faculty found themselves in a position to teach telehealth in their subject area. Compared to another study that explored the effects of time constraints on instruction [18], our study found lack of time to be a potential limitation, as reported by two participants, towards integrating a non-traditional topic such as telehealth into existing pharmacy practice curricula.

Interestingly, the faculty made suggestions in the open-ended response questionnaire to incorporate telehealth into case-based courses that leverage electronic health records. Additional suggestions were made to integrate telehealth into curricula to expose students preparing for careers with virtual provision of care. Respondents consistently shared how telehealth, as an increasingly common modality of healthcare delivery, will be essential for student pharmacists as they prepare to overcome barriers and leverage opportunities. These findings highlight opportunities to integrate tele-education into the didactic pharmacy practice curriculum and parallel other studies that emphasize the need for introductory and simulation experiential learning to improve the level of comfort with telehealth [1,3]. Pharmacy practice curricula must be promoted and developed to meet the increased demand for telehealth in the foreseeable future. 

### Limitations

This study has limitations. First, it was limited to one college of pharmacy and had a small sample size. Second, it had a low response rate. Next, telehealth was not defined in the survey. A future project could expand on this study by assessing the value of incorporating telehealth instruction (e.g., communication methods, clinical applications, examples) at other colleges/schools of pharmacy and better understanding participant level of comfort with telehealth, either solely with instruction or as a subject matter expert. Research into participant experience with telehealth either as a patient or provider would be valuable in future research. In addition, concept elicitation interviews to ensure that the instrument contains the most appropriate content and/or a Delphi panel to help ensure that potential participants clearly understand the items would be valuable for any instrument used in future research. A new survey administered to other healthcare professional programs and various institutions exploring best practice methods to incorporate telehealth into the classroom is warranted to increase the generalizability of the findings.

## 5. Conclusions

This exploratory survey found that faculty perceptions and opinions of telehealth integration into Doctor of Pharmacy training programs at one college of pharmacy was positive. Future consideration towards incorporation of telehealth into the curricula at other institutions is warranted. 

## Figures and Tables

**Table 1 pharmacy-11-00110-t001:** Demographic characteristics of study participants.

Variable	N (%)
Length of Time as a Faculty Member at Any College of Pharmacy	
<5 years	6 (35.3%)
6–10 years	3 (17.7%)
11–20 years	8 (47.0%)
Length of time as a faculty member at the College of Pharmacy	
<5 years	7 (41.2%)
6–10 years	4 (23.5%)
11–20 years	6 (35.3%)
Pharmacy degree	
Yes	14 (82.3%)
No	3 (17.7%)
Sex	
Female	12 (70.6%)
Male	5 (29.4%)
Have you received any type of telehealth service?	
Yes	12 (67%)
No	6 (33%)

One respondent did not provide demographic data.

**Table 2 pharmacy-11-00110-t002:** Participants’ responses about telehealth in the PharmD curriculum.

Variable	Result
Awareness	
How aware are you about telehealth? Median (IQR)	4.0 (0.8)
How aware are you about the applications of telehealth? Median (IQR)	3.0 (1.0)
How aware are you about the benefits of telehealth? Median (IQR)	4.0 (0.0)
How aware are you about the barriers of telehealth? Median (IQR)	4.0 (1.0)
Perception	
How positive is your perception of telehealth in general? Mean (SD)	8.1 ± 1.5
How much value do you see in telehealth service? Mean (SD)	8.6 ± 1.6
How much value do you see in the incorporation of telehealth instruction into the pharmacy practice curriculum? Mean (SD)	7.7 ± 2.7
Level of comfort	
How comfortable do you feel discussing telehealth communication? Median (IQR)	4.0 (0.5)
How comfortable do you feel discussing telehealth applications? Median (IQR)	2.5 (2.0)
How comfortable do you feel discussing the benefits of telehealth? Median (IQR)	4.0 (0.5)
How comfortable do you feel discussing telehealth barriers? Median (IQR)	4.0 (0.5)
Telehealth Instruction	
Do you teach a subject area in which you can discuss telehealth? N (%)	
Yes	12 (67%)
No	6 (33%)
Do you plan to or have you already incorporated telehealth instruction into your courses? N (%)	
Yes	8 (44)
No response	6 (33)
No	4 (22)
What is the reason why you do not teach telehealth in your class/subject area? N (%)	
I do not think it is important.	1 (25%)
I do not have time to teach it.	2 (50%)
I had not thought about incorporating it specifically, but I can.	1 (25%)

IQR = interquartile range, SD = standard deviation. Awareness items were scored on a 4-point scale: 1 = not aware, 2 = somewhat aware, 3 = aware, 4 = very aware. Perception items were scored on a 10-point scale: 0 = not very valuable, 10 = extremely valuable. Comfort items were scored on a 4-point scale: 1 = not at all comfortable, 2 = somewhat comfortable, 3 = comfortable, 4 = very comfortable.

## Data Availability

Data are unavailable due to privacy restrictions.

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
