# Peer review of "Telehealth Integration into Pharmacy Practice Curricula: An Exploratory Survey of Faculty Perception"

_pharmacy, 2023, doi:10.3390/pharmacy11040110_

Round 1

Reviewer 1 Report

Thank you for the opportunity to review your manuscript! I believe that the incorporation of telehealth (and digital health) into the pharmacy curriculum is an important and timely topic. I appreciate the author’s attempt to characterize the perceptions of telehealth among their faculty. Please see the suggestions below for possible improvements. Recognizing that this is a brief report, it may not be feasible to re-design methods but I think it is important to ensure the most robust methodology is utilized if possible to ensure meaningful measurement of outcomes.

Title – Consider revising the title as I thought this would provide more details on how telehealth has been integrated into the pharmacy curriculum and less about the faculty perception.

Intro

-          When discussing the shift in pharmacy school delivery during the pandemic, I think it will be important to weight the need to be virtual for social distancing purposes versus the need to intentionally teach methods to communicate when conducting telehealth services. These are two very different concepts.

-          Consider further explanation of the AACP task force recommendation for digital health. I believe that digital health is a broader term and could incorporate many other elements besides telehealth. So, it is likely that digital health was a curriculum need outside of the pandemic demonstrating an important need for telehealth. So still both are now clearly needed but just being clear that it is likely due to the digital age with many technology advances as well as the circumstances from the pandemic.

Methods

-          Did inclusion criteria consider faculty member expertise and if they had pharmacy practice experience to be able to determine the placement of telehealth concepts?

-          Provide details on the literature search (what were criteria, what studies were identified and utilized)

-          Provide details on how the researchers extracted items from a published checklist. Why was this checklist chosen? What details were considered?

-          For the questionnaire, did you define telehealth for the participants before they began answering the questions?

-          For the comfort questions, was there any guidance on if this was just comfort in general discussion (like in the hallway talking about experience) or was this more geared to if you would feel comfortable serving as a content expert (delivering education to students). I could see how there would be different ways of viewing these questions.

-          For the instruction portion, did you provide guidance on what it would mean to cover telehealth. Would it be communication methods specific for telehealth, clinical applications that could be done in telehealth environment, examples from telehealth experiences, etc… Having telehealth instruction could mean a lot of different things.

Results

-          Although this is an exploratory study, the response rate is pretty low, especially for an internal survey.

-          How do the demographics compare to the faculty cohort that was sent the survey. Does the respondent characteristics match what you would anticipate based on the population?

-          Table 2 – consider including the potential range in the categories (1 to 4 for awareness vs. 1 to 10 for perception) so that it is clear that these are on different scales.

-          It would be nice to know for those that responded as “yes” plans to incorporate or already incorporate telehealth how they are doing this and in what class. This may be helpful for other faculty within the institution to know but also for external faculty within the academy.

-          Since this is an exploratory survey, you could consider reporting statistics such as average response time.

-          It may be valuable to do a further validation step to see if participants understand the questions the way they are written such as a focus group or a delphi method. This may help validate the questions before distributing this survey to a broader group.

Discussion

-          Although I appreciate the relationship between these outcomes and those of healthcare provider’s perception on telehealth for care delivery, I would be careful on making a direct comparison as this survey assessed perception more from a patient perspective (how they viewed telehealth as a patient) not necessarily from serving as the healthcare provider. Unless perhaps you can separate responses for those that have provided telehealth services from those that have not provided those types of services.

-          The discussion says that most faculty felt comfortable teaching about telehealth but the survey says “discuss telehealth”. I think these words are two very different things and I would be careful on the interpretation of this section. An alternative method may be to collect data on faculty that have received formal training on telehealth communication or digital health applications. Or perhaps an assessment of reading or research that they have done on this topic to demonstrate if they would be prepared to teach this type of content.

-          I really like the suggestion to incorporate telehealth competencies that are specific to pharmacy and perhaps to use the medical colleges format, but I think the connection to the study results needs to be more fully formed. How did the study results support this. (although it really is a great idea!!!).

-          I do believe that time is likely a big factor into faculty lack of incorporation of telehealth components but there were only two faculty from the survey that noted that as the reason. So, I think it is important to add some more speculation to this discussion. While it is very possible, it is hard to say that the “most” faculty since it was such a small number.

-          The discussion of the open-ended suggestions is important. I actually think this is the biggest take-away rather than the quantitative findings. I would consider re-designing the methods of this study to do a deeper dive into faculty suggestions, either through focus groups or delphi methods.

-          For the limitations, please consider adding details on why you believe such a low response rate. I unfortunately think that is a very large limitation of the study and therefore it is hard to extrapolate these findings. It would also be important to add some validation to the questions to ensure that they are gathering the information that you are intending to gather.

The manuscript was well written and easy to follow. Minor edits during the final production phase would be reasonable.

Reviewer 2 Report

This is a very well written manuscript describing an exploratory study on perceptions of pharmacy faculty regarding integration of telehealth content to pharmacy curriculum. While acknowledging that this is intended to be an exploratory study, the results seem somewhat generalized and there is a very small sample size of respondents that are all from one university. However, even with the limitations, the results of this study demonstrate that further research is needed to identify barriers to incorporating telehealth education to pharmacy curriculum and to identify processes to guide smooth integration.

The most significant edit needed to this manuscript is that more description (with included citations-see further comment below) should be added to explain how the survey was created. Otherwise, there are a few minor edits listed below. The paper reads very well with minimal noted grammatical errors.

Page 1-line 40-appears to be an extra space after citation in brackets

Page 1-line 41-it appears there is a citation not in brackets

Page 2-Methods-Exploratory Questionnaire section-states that a literature review helped shape the current study survey but there are no citations for the literature that was utilized for this; please include more description as well as the citations for the literature utilized to create current survey

Page 2-need citation for the ‘published checklist’ used to help create current study survey

Page 3- line 129-citation is not in brackets

Page 3-methods-how were data analyzed, was software utilized?

Page 3-data collection-line 128; should be data were collected…(not was)

Page 5-Table 2-it would be helpful to have the range of scores possible within each of the three categories (awareness, perception, level of comfort) since they are different for the different categories

Round 2

Reviewer 1 Report

Thank you for all the revisions! Your edits addressed all the suggestions!